# Light quality-regulated anthocyanin biosynthesis in *Lilium leichtlinii* subsp. *maximowiczii* bulbs: A multi-omics perspective

**Jinxing Xu**[1⚬‡], **Shuai Li**[1⚬‡], **Chen Wang**[1], **Bingyan Liu**[1], **Qi Wu**[1], **Yanli Ma**[1], **Xinzhu Dai**[3], **Yuanhang Zhou**[3], **Xinhai Yu**[2]*, **Tianliang Wang**[1]*

**1** College of Landscape Architecture, Changchun University, Changchun, China, **2** Jilin Institute of Biology, Changchun, China, **3** Changchun Academy of Forestry Science, Changchun, China

⚬ Jinxing Xu and Shuai Li contributed equally to this work.
‡ These authors share first authorship.
* 15164357527@163.com (XY); wang.tl@hotmail.com (TW)

## Abstract

This study elucidated the molecular mechanisms governing light spectrum-regulated coloration and secondary metabolism in *Lilium leichtlinii* subsp. *maximowiczii* bulbs during postharvest storage. Chromatic profiling demonstrated that continuous white (WLc) and blue light (BLc) treatments synergistically induced violet-red pigmentation ($\Delta E = 55.2$ and $52.1$ at day 12) through rapid anthocyanin biosynthesis (up to 79.39-fold accumulation), while continuous red light (RLc) promoted yellowish-brown discoloration via flavonoid pathway activation. Multi-omics integration showed WLc-mediated reprogramming of light signaling cascades and metabolic networks. Metabolomic analysis identified 177 differentially accumulated metabolites, with cyanidin-3-rutinoside (keracyanin, Fold Change [FC] = 316.53) as the predominant pigment. Transcriptomic profiling uncovered coordinated upregulation of phenylpropanoid (*PAL*, *C4H*, *CHS*) and anthocyanin pathway genes (*F3H*, *DFR*, *ANS*, *3GT*), with 15,823 differentially expressed genes enriched in flavonoid biosynthesis (KEGG, $p < 0.01$). Enzyme kinetic analysis highlighted hierarchical gene activation patterns: upstream pathway genes (*CHS*, *F3H*) exhibited peak expression at day 3, while downstream genes (*DFR*, *ANS*) gradually increased to synchronize pigment production. Phylogenetic validation confirmed >95% conservation of catalytic domains (2-OG oxygenase in LlF3H/LlANS; MGT motif in Ll3GT) across the Liliaceae family. Environmental factors such as storage temperature and packaging permeability were shown to modulate these light-responsive pathways. This work establishes a comprehensive spectral regulation network model, providing mechanistic insights for lily color breeding programs and spectrum-controlled postharvest technology development.

**Data availability statement:** The minimal dataset needed to replicate this study is available at https://doi.org/10.6084/m9.figshare.32054361.

**Funding:** This study was supported by Science and Technology Development Program of Jilin Province (2024JB427L35).

**Competing interests:** The authors have declared that no competing interests exist.

## 1. Introduction

*Lilium*, a genus of perennial herbaceous flowering plants indigenous to the temperate Northern Hemisphere, encompasses approximately 115 species, with 55 species distributed across China. These plants hold substantial commercial significance in floriculture due to their distinctive floral morphology, vivid hues derived from specialized chromoplast pigments (carotenoids and anthocyanins), and pronounced floral fragrance produced by volatile terpenoid compounds. Additionally, their bulbous organs demonstrate cosmeceutical properties, including skin moisturization, neuro-anxiolytic effects, and antitussive efficacy.

Amid global shifts toward preventive healthcare, market demand for lilies has surged exponentially. In China, the nutraceutical sector derived from Lilium species demonstrated robust expansion, with market volume increasing from ¥380 million in 2019 to ¥454 million in 2023, reflecting a compound annual growth rate (CAGR) of 4.52%. Projections based on headstream analytics anticipate accelerated growth, potentially reaching a 5.27% CAGR by 2028. However, during postharvest storage and transportation, lily bulbs are highly susceptible to mechanical damage or environmental stressors, leading to rot and color alterations [1]. These alterations not only compromise visual quality but may also reduce their nutritional value and market competitiveness.

Color deterioration in lily bulbs during postharvest storage occurs through two primary mechanisms: enzymatic browning and anthocyanin-dependent violet-red pigmentation [2]. Zhao et al. demonstrated a significant positive correlation between browning severity and malondialdehyde (MDA) content, polyphenol oxidase (PPO) activity, and secondary metabolite accumulation [3]. The violet-red pigmentation of lily bulbs is regulated by the anthocyanin biosynthesis pathway [4]. Fan et al. found that light exposure induces violet-red pigmentation in *Lilium davidii* subsp. *unicolor* (Lanzhou lily), with transcriptomic and metabolomic analyses revealing upregulated expression of flavonoid biosynthesis genes and elevated levels of epicatechin, rutin, and cyanidin 3-rutinoside [5]. Mitigation strategies such as heat treatment [6], vacuum packaging [7], low-temperature storage [8], and sulfur fumigation [9] have proven effective in reducing light-induced color changes in lily bulbs.

*Lilium leichtlinii* subsp. *maximowiczii* is the most representative edible lily variety in the Changbai Mountain region of Jilin Province, China. This newly characterized variety demonstrates key phenotypic advantages, including thick fleshy scales, excellent flavor quality, and high biomass accumulation efficiency, indicating substantial market potential. Its strategic importance extends to medicinal-edible product innovation and health industry development [10]. Current research on violet-red pigmentation in lily bulbs remains limited, predominantly focusing on cultivars such as *Tresor* and Lanzhou lilies, while *L. leichtlinii* subsp. *maximowiczii* remains understudied. Moreover, mechanistic understanding of this pigmentation is incomplete, with critical genes and regulatory networks yet to be fully elucidated. This study systematically investigates the molecular regulatory mechanism governing anthocyanin accumulation in the bulbs of *L. leichtlinii* subsp. *maximowiczii* through controlled experiments examining light quality, temperature, and packaging methods. We employ integrated approaches

combining chromaticity analysis, quantitative determination of anthocyanins, flavonoids, and total phenolics with multi-omics profiling (transcriptomics and metabolomics). The research prioritizes identification of key structural genes and regulatory factors involved in anthocyanin biosynthesis. Specifically, we aim to: characterize chromatic changes and correlate them with metabolite profiles; identify critical inducing factors; classify pigment-associated metabolites; and decode regulatory networks. The findings provide a theoretical foundation for exploring new technologies to enhance the postharvest appearance quality of *L. leichtlinii* subsp. *maximowiczii* bulbs.

## 2. Materials and methods

### 2.1 Plant materials

**2.1.1 Different light quality treatments.** This study utilized healthy and uniformly matured *Lilium leichtlinii* subsp. *maximowiczii* bulbs obtained from a certified seed farm in Zhuchengzi Town, Dehui City, Jilin Province, China (44°31′60″N,125°56′11″E). Only bulbs exhibiting consistent size, color, and maturity, with no signs of mechanical injury or pathological infection, were selected for experimentation. After gentle scale separation and thorough cleaning, bulbs were randomly divided into five groups and placed in glass Petri dishes. All samples were stored at 10 °C under controlled light conditions: continuous exposure to white light (WLc; 400–700 nm, 25 $\mu$mol·m$^{-2}$·s$^{-1}$), blue light (BLc; 450 nm, 25 $\mu$mol·m$^{-2}$·s$^{-1}$), green light (GLc; 550 nm, 25 $\mu$mol·m$^{-2}$·s$^{-1}$), red light (RLc; 660 nm, 25 $\mu$mol·m$^{-2}$·s$^{-1}$), or darkness (Dc). Samples were collected at 0, 1, 3, 6, 9, and 12 days post-storage and immediately frozen at –80 °C for subsequent analysis.

**2.1.2 Different temperature treatments.** Cleaned bulbs were evenly divided into six experimental groups using a constant-temperature incubator. (PYL-125, Tianjin Laboratory Instrument Co., Ltd., China). All groups were subjected to three temperature conditions (4 °C, 25 °C, 37 °C) with two light treatments per temperature: continuous exposure to white light (WLc) and darkness (Dc). During storage, the relative humidity was maintained at 75%±5%. Samples were collected on days 0, 3, 6, 9, and 12, and immediately stored at –80 °C.

**2.1.3 Different packaging conditions.** Bulbs were divided into four groups: two under vacuum packaging (VP; <0.01 atm) using a heat sealer and two under normal packaging (NP; ambient O$_2$). All groups were simultaneously placed under two different light treatments: Dc and WLc, resulting in a total of four experimental groups. During storage, samples were taken out at 0, 1, 3, 6, 9, and 12 days, and immediately stored at –80 °C.

### 2.2 Determination of apparent color

Colorimetric measurements of *L. leichtlinii* subsp. *maximowiczii* bulbs were conducted using a precision colorimeter (*LS170*, Shenzhen LinShang Technology Co., Ltd., China). Following standard calibration, colorimetric parameters of lily bulbs were measured at 0, 1, 3, 6, 9, and 12 days. Specifically, Lightness (L*), Red/green coordinate (a*), Yellow-blue coordinate (b*), and Total color difference (ΔE) relative to day 0 were quantified. The L* value indicates color lightness, ranging from 0 (pure black) to 100 (pure white). Positive a* values represent red chroma, with higher values corresponding to increased red intensity. Positive b* values represent a yellow chromaticity gradient, where the efficacy of yellow coloration increases with higher b* values [11]. The ΔE value denotes the color difference, calculated according to the following formula:

$$\Delta E * ab = \sqrt{(\Delta L_*)^2 + (\Delta a_*)^2 + (\Delta b_*)^2}$$

### 2.3 Analysis of secondary metabolites

**2.3.1. Anthocyanin quantification.** Anthocyanins were extracted using a solvent mixture (18% n-propanol, 1% HCl, and 81% dH$_2$O). Fresh bulb samples (1 g) were mixed with 3 mL of the extraction solution, heated at 100°C for 3 min,

and incubated overnight at room temperature in darkness. Absorbance was measured at 535 nm and 650 nm using a spectrophotometer (*i3*, Hanon Group Co., Ltd., China).

**2.3.2. Determination of Total Phenolic Content (TPC) and Total Flavonoid Content (TFC).** Modified protocols from Sun et al. [12] were employed. Briefly, freeze-dried lily bulbs were ground into powder using liquid nitrogen, and 5 mg of the sample was dissolved in 20 mL of methanol, ultrasonicated for 20 minutes, and then centrifuged at $6000 \times g$ for 5 min at 4 °C. The final methanol extract was collected for further analysis.

TPC was determined using the Folin-Ciocalteu method [13] with minor modifications. Methanol extract (50 µL) was mixed with 500 µL of 10 × diluted Folin-Ciocalteu reagent. After 5 min, 450 µL of 7.5% sodium carbonate ($Na_2CO_3$) was added, and the mixture was incubated at room temperature in the dark for 60 min. Absorbance was then measured at 760 nm using a microplate reader (Epoch2, BioTek Instruments, Inc., USA).

TFC was measured using a colorimetric assay [14]. Methanol extract (100 µL) was treated sequentially with 15 µL of 5% sodium nitrite ($NaNO_2$) (5 min), 15 µL of 5% $AlCl_3$ (6 min), and 100 µL of 1 M NaOH (30 min incubation). Absorbance was measured at 510 nm using a microplate reader (Epoch2, BioTek Instruments, Inc., USA).

## 2.4 Metabolite extraction and analysis

Samples from untreated controls (Ctrl) and WLc-exposed bulbs at day 6 (WL6) were analyzed for metabolic composition. The extraction, detection, and quantitative analysis of metabolites in the samples were entrusted to Wuhan MetWare Biotechnology Co., Ltd. (Wuhan, China). Briefly, freeze-dried lily bulb samples were weighed and extracted overnight at 4 °C with 1.0 mL of 70% methanol. The extracts were processed and analyzed via LC-MS/MS. Metabolites were annotated against the MetWare database (MWDB) and quantified using multiple reaction monitoring (MRM) with Analyst 1.6.1 software. Multivariate analysis included orthogonal partial least squares-discriminant analysis (OPLS-DA) and principal component analysis (PCA) to validate the variability and reliability of metabolites across samples. Differentially accumulated metabolites (DAMs) were screened and selected by variable importance in projection (VIP ≥ 1) and statistical significance ($p < 0.05$). Pathway enrichment was performed using the Kyoto Encyclopedia of Genes and Genomes (KEGG) database, with three technical replicates per sample to ensure reproducibility and reliability.

## 2.5 Transcriptomics

RNA-Seq libraries were constructed from Ctrl and WL6-treated bulbs. Total RNA was extracted from *L. leichtlinii* subsp. *maximowiczii* bulb samples using the centrifugal column-based Total RNA Extraction Kit (DP419, Tiangen Biotech Co., Ltd., Beijing, China) following the manufacturer's instructions. RNA degradation and DNA contamination were assessed by RNase-free agarose gel electrophoresis. The integrity and purity of the RNA samples were evaluated using the Agilent 2100 Bioanalyzer system (Agilent 2100, Agilent Technologies, USA) and the Nano Drop One spectrophotometer (Nanodrop One, Thermo Fisher Scientific, USA), respectively. The extracted RNA from the lily bulbs was utilized to construct RNA-Seq libraries. Samples undergoing various treatments were analyzed in triplicate. The prepared cDNA libraries were sequenced on the Illumina HiSeq 2500 platform by MetWare Biotechnology Co., Ltd. (Wuhan, China). Differentially expressed genes (DEGs) were identified with an absolute fold change (FC) > 2.0, $p < 0.05$. Functional enrichment was performed with KOBAS 3.0 [15] for Gene Ontology (GO) [16] terms and KEGG pathways, while transcription factors were predicted using ITAK software [17]. All treatments were analyzed in biological triplicates to ensure statistical reliability.

## 2.6 Quantitative real-time PCR (qRT-PCR) Validation

To validate the accuracy of the RNA-Seq data, 25 differentially expressed candidate genes encoding key enzymes potentially involved in anthocyanin biosynthesis were selected for qRT-PCR analysis. The specific primers for these genes are listed in S1 Table 1. Total RNA was extracted from lily samples subjected to different treatments (Ctrl and WL6) using the

Total RNA Extraction Kit (DP419, Tiangen Biotech Co., Ltd., Beijing, China). The RNA samples used were the same batch as those employed for transcriptome sequencing. Subsequently, reverse transcription was performed using Innovagene's TRUEscript RT MasterMix (AR121-Mix, Hunan Innovagene Biotechnology Co., Ltd., China) according to the manufacturer's instructions to synthesize first-strand cDNA. Quantitative real-time PCR (qPCR) analysis was then carried out using Innovagene's 2×Taq SYBR Green qPCR Mix (With ROX) kit (SQ101−01, Hunan Innovagene Biotechnology Co., Ltd., China) and an Applied Biosystems 7500 Fast System (Thermo Fisher Scientific, USA). Actin and 18S were used as reference genes. The qRT-PCR procedure was carried out under the following conditions: 94 °C for 2 minutes, followed by 40 cycles of 94 °C for 10 seconds, 60 °C for 30 seconds, and 72 °C for 20 seconds. Relative gene expression was calculated via the $2^{-\Delta\Delta Ct}$ method, with three technical replicates per sample to ensure reproducibility and reliability.

### 2.7 Bioinformatics analysis of key enzyme sequences

Full-length cDNA sequences of LIF3H, LIANS, and Ll3GT were aligned against NCBI (http://www.ncbi.nlm.nih.gov/) reference sequences using DNAMAN v9.0. ProtParam (https://web.expasy.org/protparam/) was employed to predict the amino acid composition, isoelectric point, relative molecular weight, and other physicochemical properties of the protein. Phylogenetic tree analysis was performed using the MEGA program.

## 3. Results and analysis

### 3.1 Light quality and storage duration drive differential chromatic transitions in bulbs

As illustrated in Fig 1A, bulb coloration in *L. leichtlinii* subsp. *maximowiczii* underwent significant chromatic transitions in response to varying photostorage conditions. WLc, BLc, and GLc treatments significantly enhanced violet-red pigmentation. Notably, WLc/BLc treatments triggered rapid L* decline (ΔL*=−13.0 to −9.78 within 3 days), culminating in minimal values of 35.5 (WLc) and 37.0 (BLc) by day 12, representing 57.6% and 56.9% reductions versus initial baseline ($p < 0.05$; S1 Fig A). Concurrently, a* of bulbs in the WLc and BLc groups exhibited significant increases of 9.7 and 7.5 units on day 3, which were markedly higher than those observed in other treatment groups (S1 Fig B). RLc exposure promoted yellowish-brown pigmentation, with b* values (yellow-blue chromaticity) increasing by 33.3% from 10.8 to 14.4, while L* remained elevated (65.2) compared to WLc/BLc/GLc groups (S1 Fig C). In contrast, color changes under Dc conditions were minimal, with the L* value only decreasing to 79.3, and a* consistently remaining negative. The ΔE revealed that after 12 days, the WLc and BLc treatments reached 55.2 and 52.1, respectively, significantly higher than other groups, indicating that BLc and WLc had the strongest regulatory effects on long-term storage-induced color changes (S1 Fig D).

Collectively, light quality and storage duration cooperatively influenced bulb color evolution: white and BLc significantly promoted violet-red pigmentation, RLc induced yellowish-brown pigmentation, while darkness decelerated these oxidative processes, preserving initial color integrity.

### 3.2 Light-specific metabolic reprogramming in postharvest lily bulbs

To investigate the regulatory effects of light quality on metabolites in *L. leichtlinii* subsp. *maximowiczii* bulbs during storage, this study analyzed the dynamic changes in primary and secondary metabolites. The results showed that primary metabolites (total sugars, sucrose, starch, and total amino acids) declined steadily over the 12-day storage period, indicating the gradual consumption of energy reserve substances (S2 Fig). In contrast, secondary metabolites exhibited significant light-dependent accumulation: WLc, BLc, and GLc treatments triggered a rapid synthesis of anthocyanins within 3 days, reaching 48.07-, 79.39-, and 32.62-fold increases over initial levels, respectively (Fig 1B). By day 12, WLc-treated bulbs achieved peak anthocyanin content (2.93 mg/g), 10.37-fold higher than RLc groups, while GLc group maintained 4.39-fold elevation. Notably, BLc treatment induced a significant increase in total phenolics early in storage (day 1), reaching 24.28 mg/g by day 12, which was 8% higher than in the WLc group. Meanwhile, flavonoids in the WLc group accumulated continuously, reaching 1.96 mg/g, which showed no significant difference from the BLc group (Fig 1C).

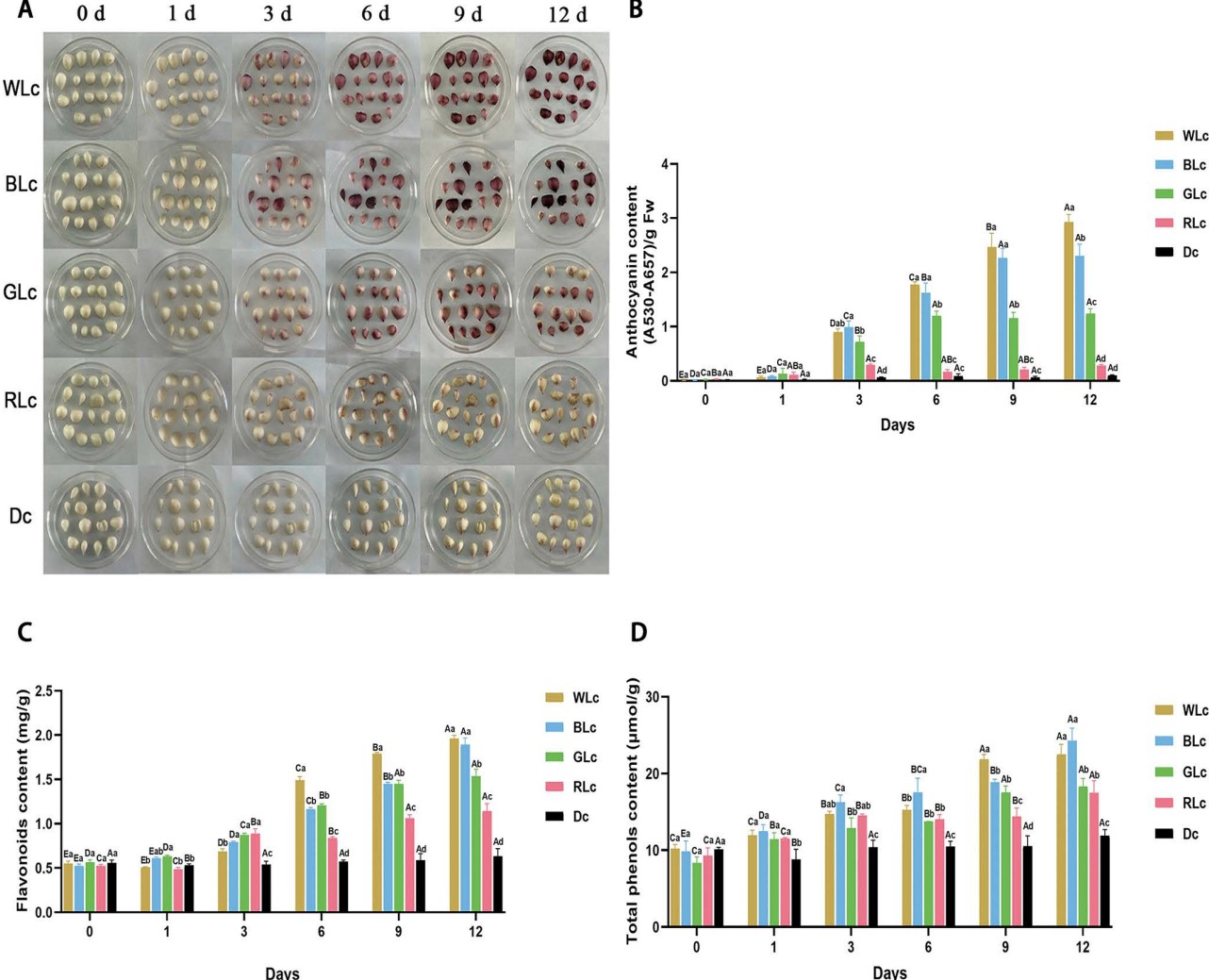

**Fig 1. Phenotypic responses and dynamics of secondary metabolites in *Lilium leichtlinii* subsp. *maximowiczii* bulbs to spectral treatments.**
**(A)** Bulb appearance under different light qualities. **(B)** Anthocyanins [(A530-A657)/g FW]. **(C)** Flavonoids (mg/g). **(D)** Total phenolics (μmol/g). Capital letters (A–E) indicate significant differences across storage days (0-12 d) under identical light; lowercase letters (a–d) indicate significant differences among light treatments (WLc, BLc, GLc, RLc, Dc) at identical storage time ($p < 0.05$ by two-way ANOVA with Tukey's test). Data presented as mean±SD (n=3). WLc: white light; BLc: blue light; RLc: red light; GLc: green light; Dc: dark control.

RLc treatment exhibited a unique regulatory pattern: anthocyanins peaked transiently on day 3, while flavonoids and total phenolics accumulated slowly in the later storage period. Dc suppressed metabolic activation, with no significant secondary metabolite variation. Time-course analysis revealed BLc-driven early anthocyanin synthesis, with a 79-fold increase relative to Dc by day 3, while WLc promoted sustained accumulation, reaching a 48-fold increase by day 12. GLc selectively enhanced anthocyanin biosynthesis without activating total phenolics. Mechanistically, WLc and BLc synergistically promoted total phenolic accumulation (Fig 1D), RLc preferentially activated flavonoid pathways, and GLc specifically modulated anthocyanin biosynthesis. These findings delineate the differential regulatory mechanisms of spectral components on bulb coloration and stress-resistant compound synthesis, providing metabolic insights for postharvest quality management.

### 3.3 Temperature optima divergently regulate light-dependent pigmentation and secondary metabolism

The bulb coloration and secondary metabolism of *L. leichtlinii* subsp. *maximowiczii* during storage exhibit distinct temperature-light interactions. WLc induced pronounced violet-red pigmentation, while Dc suppressed this response. Temperature critically modulated pigmentation kinetics: under WLc, bulbs stored at 4 °C and 25 °C showed progressive violet-red intensification (evident by day 3 at 25 °C), whereas 37 °C storage abolished this response (Figs 2A and 2B). Dark storage at 4 °C preserved bulb whiteness (highest L* value) and minimized color change (smallest ΔE value), representing optimal conditions for color stability. Colorimetric analysis revealed that WLc accelerated L* value decline and significantly increased a* values (red shift), while decreasing b* values (except at 37 °C). In contrast, Dc elevated b* values. All treatments showed rising ΔE trends (S3 Fig).

Anthocyanin accumulation demonstrated strict light-temperature dependence: WLc 4 °C and 25 °C significantly enhanced synthesis to 18.21- and 28.82-fold initial levels by day 6, reaching 116.40- and 63.91-fold higher than the 37 °C group by day 12 ($p < 0.05$; Fig 2C, 2D), while 37 °C strongly suppressed it. Dc suppressed anthocyanins below detection limits.

Flavonoids (Fig 2E and 2F) and total phenolics (Fig 2G and 2H) increased throughout storage but were less light-dependent. High temperature (37 °C) strongly induced their accumulation. On day 12 under WLc, flavonoid content reached 0.98, 1.40, and 1.84 mg/g at 4, 25, and 37 °C, respectively (Dc: 0.93, 1.76, and 1.93 mg/g). Total phenolics under WLc were 17.61, 28.71, and 27.22 µmol/g at corresponding temperatures, with Dc groups showing 2.35- and 3.18-fold increases over initial levels at 25 and 37 °C. These results demonstrate pathway-specific temperature optima: WLc and low temperature (4 °C) preferentially promote anthocyanin biosynthesis, while elevated temperature (37 °C) activates flavonoid and phenolic pathways.

### 3.4 Packaging permeability dictates light-induced metabolic pathways during storage

This study investigated the interactive effects of packaging permeability (VP compared to NP) and light exposure (WLc compared to Dc) on color development and secondary metabolite accumulation in *L. leichtlinii* subsp. *maximowiczii* bulbs during storage (Figu 3A and 3B). Under WLc, NP-packaged bulbs developed pronounced violet-red pigmentation (evident by day 6; a* value reached maximum 13.3 on day 12, representing a 21.1-fold increase from day 0), accompanied by decreased L* values and increased ΔE (S4 Fig). This response was significantly suppressed by VP packaging. In contrast, neither packaging type induced pigmentation under Dc, with no significant differences in color parameters. Anthocyanin accumulation was exclusively observed in WLc + NP bulbs (14.39-fold initial levels by day 12; Fig 3C and 3D). Flavonoids were consistently higher in NP-treated bulbs compared to VP groups under both WLc (1.41-fold) and Dc (1.70-fold) conditions (Fig 3E and 3F). Total phenolics exceeded initial levels in all treatments by day 6; by day 12, NP groups maintained higher levels than VP (WLc: 1.56-fold; Dc: 1.95-fold; Fig 3G and 3H). These results indicated that VP packaging stabilized bulb color and inhibited light-induced anthocyanin biosynthesis, while NP packaging promoted light-independent flavonoid/phenolic accumulation. Light exposure remained the primary driver of violet-red pigmentation, with packaging permeability acting as a regulatory modulator for these metabolic pathways.

### 3.5 Systems metabolomics identifies cyanidin derivatives as key pigmentation effectors of photoinduced bulbs

Principal component analysis (PCA) of *L. leichtlinii* subsp. *maximowiczii* bulbs from Ctrl (pre-treatment) and WL6 (6-day WLc) groups provided a systematic overview of light-induced metabolic reprogramming (Fig 4A). The PCA model showed that Principal components 1 (PC1) and 2 (PC2) accounted for 84.60% and 6.87% of total variance, respectively, collectively explaining 91.47% of metabolic variation. This clear separation between Ctrl and WL6 groups confirmed significant light-induced alterations in bulb metabolism. Metabolomic profiling identified 252 flavonoid metabolites, with flavonols (96 compounds) and flavones (44 compounds) as the predominant subclasses, followed by anthocyanins (22 compounds)

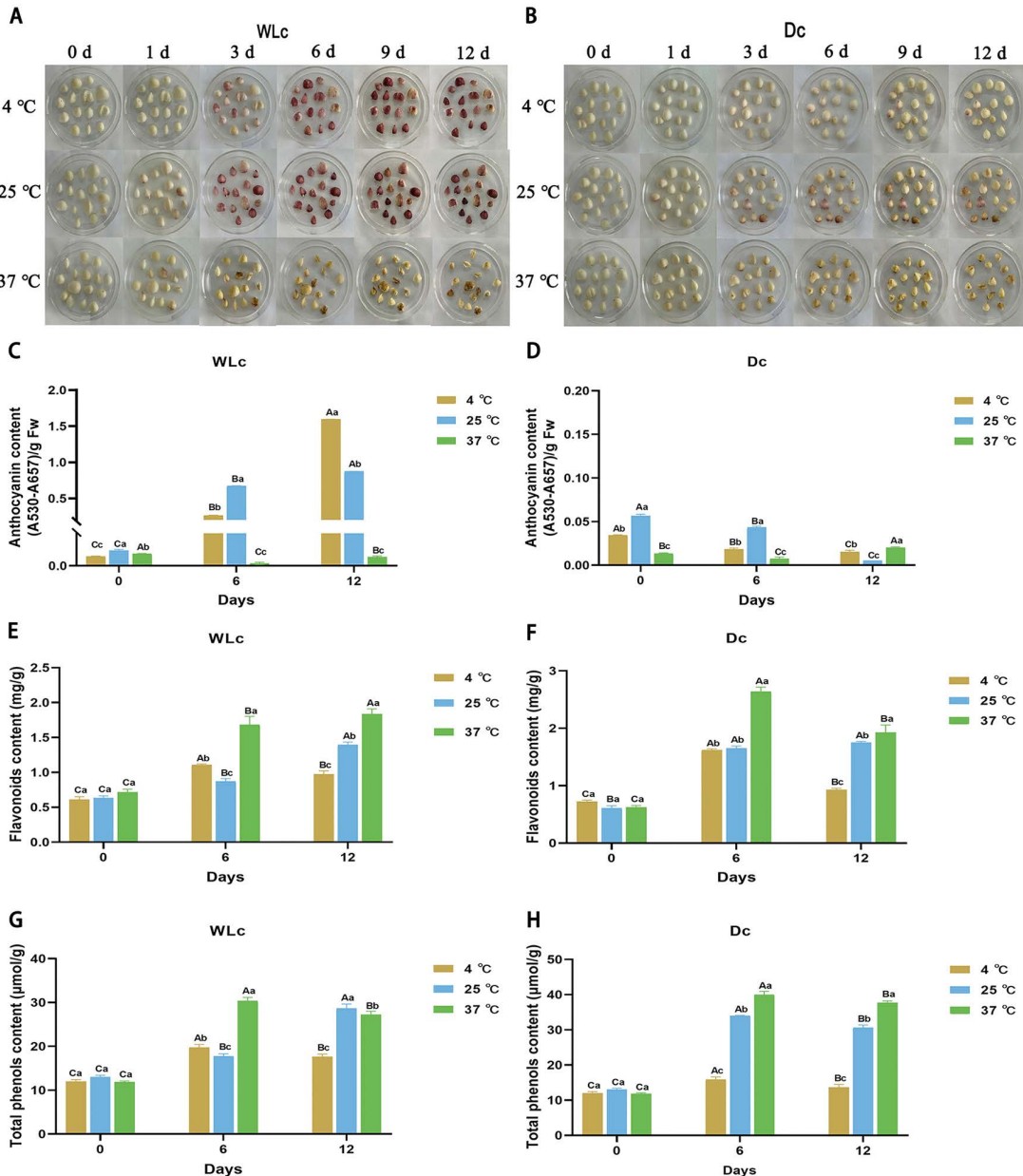

**Fig 2. Thermo-optically regulated phenotypic and secondary metabolite dynamics in *Lilium leichtlinii* subsp. *maximowiczii* bulbs during storage. (A)** Visual appearance under white light (WLc) and (B) continuous darkness (Dc) at 4 °C, 25 °C, and 37 °C. **(C, D)** Anthocyanin content [(A530-A657)/g FW]. **(E, F)** Flavonoid content (mg/g). **(G, H)** Total phenolic content (µmol/g). Capital letters **(A-C)**: Significant differences across storage days (0, 6, 12 d) within same temperature; lowercase letters **(a-c)**: Significant differences among temperatures (4 °C, 25 °C, 37 °C) at identical storage time ($p < 0.05$ by two-way ANOVA with Tukey's test). Data presented as mean ± SD (n = 3). WLc: white light; Dc: dark control.

and dihydroflavonoids (26 compounds) (S1 Table 2). Volcano plot analysis revealed 177 differentially accumulated metabolites (DAMs) ($|\log_2 FC| \geq 1$, $p < 0.05$) in WL6, with 173 upregulated and 4 downregulated (Fig 4B). These findings underscore the potent stimulatory effect of WLc on flavonoid biosynthesis, as evidenced by the predominant upregulation of flavonoid-class metabolites.

 

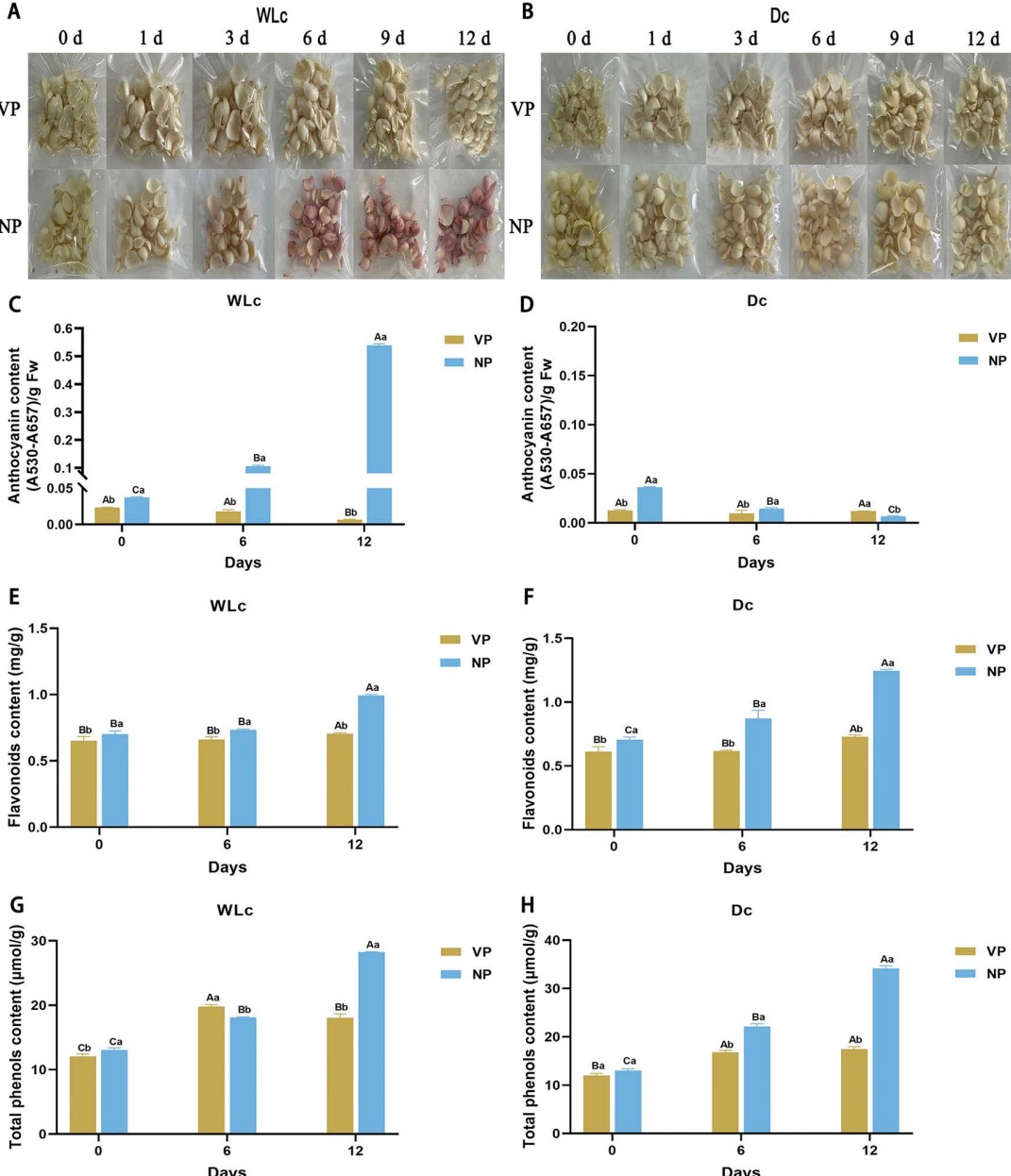

**Fig 3. Phenotypic and secondary metabolite responses of *Lilium leichtlinii* subsp. *maximowiczii* bulbs to packaging treatments during storage in light and dark conditions. (A)** Visual appearance under white light (WLc) and **(B)** continuous darkness (Dc) with vacuum packaging (VP) vs. normal packaging (NP). **(C, D)** Anthocyanin content [(A530-A657)/g FW]. **(E, F)** Flavonoid content (mg/g). **(G, H)** Total phenolic content (μmol/g). Capital letters **(A–C)**: Significant differences across storage days (0, 6, 12 d) within same packaging; lowercase letters **(a–b)**: Differences among packaging types (VP, NP) at identical storage time ($p < 0.05$ by two-way ANOVA with Tukey's test). Data are presented as mean ± SD (n = 3). Packaging: VP (vacuum packaging); NP (normal packaging). WLc: white light; Dc: dark control.

KEGG enrichment analysis showed significant enrichment of differential metabolites in secondary metabolic pathways ($p < 0.01$), including phenylpropanoids biosynthesis (Rich Factor, RF = 0.95), flavonoid biosynthesis (RF = 0.89), anthocyanin biosynthesis (RF = 0.88), flavone and flavonol biosynthesis (RF = 0.85), and flavonoid degradation (RF = 0.80)

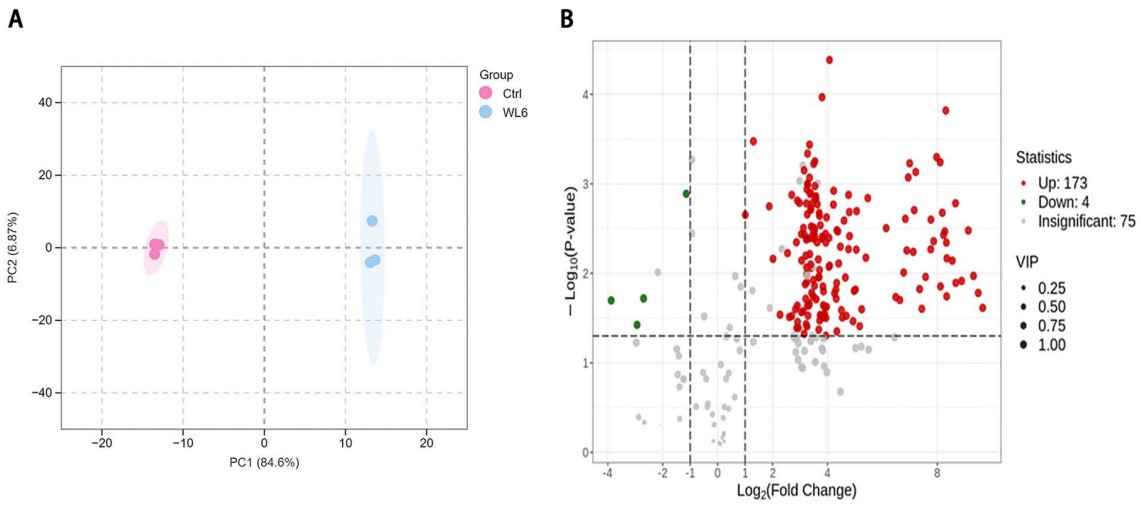

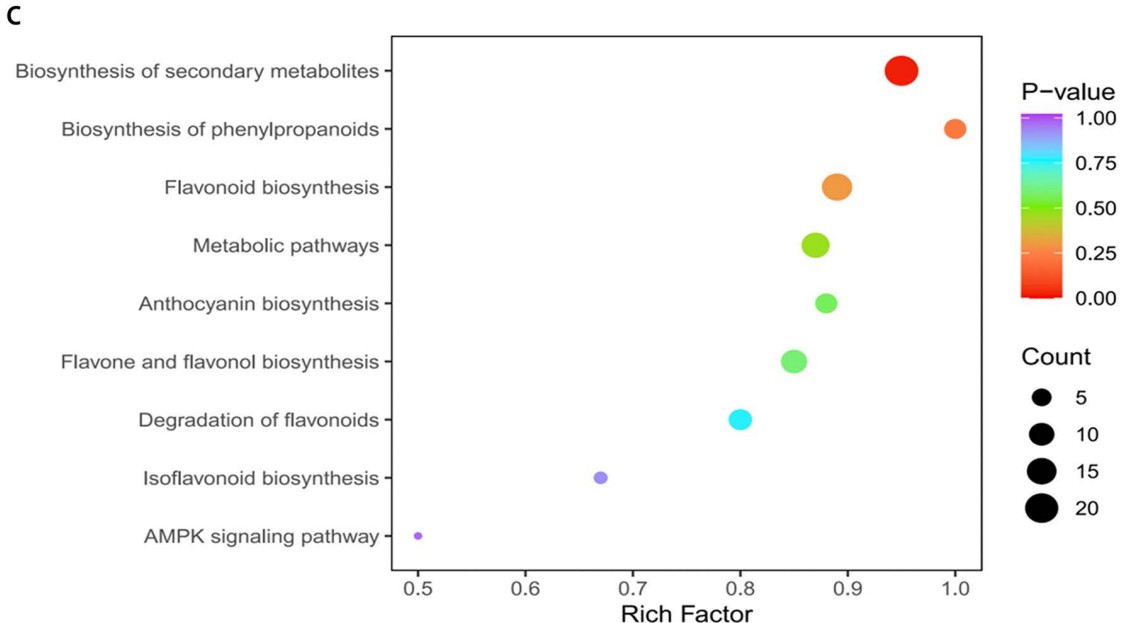

**Fig 4. Metabolomic profiling of flavonoid dynamics in *Lilium leichtlinii* subsp. *maximowiczii* bulbs under white light treatment. (A)** Principal component analysis (PCA) of Ctrl vs. WL6 groups. **(B)** Volcano plot of differentially accumulated metabolites (DAMs) (Fold change > 1.0, $p < 0.05$, VIP ≥ 1.0). **(C)** KEGG enrichment of DAMs. Ctrl: day 0; WL6: 6-day white light.

(Fig 4C). Among metabolites with $\log_2 FC \geq 4.79$, significant upregulation occurred in flavonols (30 compounds), flavones (4 compounds), dihydroflavonoids (6 compounds), other flavonoids (2 compounds), and anthocyanins (6 compounds). Notably, flavonol glycosides such as isorhamnetin-3-sophoroside ($C_{28}H_{32}O_{17}$, FC = 803.48) and quercetin-3-glucoside (isoquercitrin, FC = 717.65) exhibited the highest accumulation levels. Among flavone glycosides, madreselvin A ($C_{28}H_{32}O_{17}$, FC = 636.91) showed the highest expression. In dihydroflavonoids, hesperetin glycosides displayed higher expression levels. Anthocyanin metabolites exhibited specific accumulation patterns: WLc treatment induced significant upregulation of six anthocyanin derivatives, with cyanidin-3-rutinoside (keracyanin, FC = 316.53) showing the greatest increase, followed

by cyanidin-3-galactoside (FC = 142.63) and cyanidin-3-glucoside (FC = 139.20). Delphinidin and petunidin derivatives also exhibited 44.20- to 70.31-fold enrichment (S1 Table 3). Cyanidin-type anthocyanins demonstrated the strongest light-induced accumulation, suggesting their role as primary pigments underlying violet-red bulb pigmentation.

Further analysis indicated that WLc treatment drove rapid anthocyanin accumulation by concomitantly upregulating upstream phenylpropanoid metabolism and downstream flavonoid biosynthesis. The enrichment of "flavonoid degradation" (RF = 0.80) suggested metabolic homeostasis may be maintained through degrading enzymes (e.g., peroxidases), potentially mitigating cytotoxicity from metabolite overaccumulation. This study provided key metabolomic insights into light-mediated bulb coloration mechanisms, demonstrating that WLc dynamically regulates cyanidin derivative synthesis while maintaining metabolomic equilibrium through coordinated pathway regulation.

### 3.6  Transcriptome assembly reveals coordinated transcriptional activation of anthocyanin biosynthetic machinery under WLc

De novo transcriptome assembly of *L. leichtlinii* subsp. *maximowiczii* bulbs was performed using Trinity software [18], yielding 62,523 unigenes with an average length of 1,199 bp and an $N_{50}$ of 1,603 bp, indicating high assembly completeness (S1 Table 4). Functional annotation via DIAMOND BLASTX and HMMER identified 45,033 annotated unigenes (72.03% of total) in at least one database. Among these, the NCBI non-redundant database covered 70.40% (44,018 unigenes), while KEGG, Swiss-Prot, and GO annotations covered 52.99%, 54.38%, and 60.25%, respectively (S1 Table 5). This comprehensive annotation facilitated subsequent metabolic pathway analysis.

Sample correlation analysis revealed Pearson correlation coefficients exceeding 0.99 ($p < 0.05$) across all biological replicates (Fig 5A). Combined with PCA, this confirmed robust intra-group reproducibility and distinct metabolic separation between Ctrl and WL6 groups (Fig 5B). Differential expression screening ($|\log_2 FC| \geq 1$, $p < 0.05$) identified 15,823 DEGs, comprising 7,825 upregulated and 7,998 downregulated genes (Fig 5C). GO enrichment analysis demonstrated significant enrichment ($p < 0.05$) of anthocyanin biosynthesis-related genes, including anthocyanin-containing compound *biosynthetic process* (GO:0009718) and *flavonoid biosynthetic process* (GO:0009813), as well as core catalytic activities such as *chalcone synthase activity* (GO:0102128) (Fig 6A).

KEGG pathway enrichment analysis further revealed significant enrichment of DEGs in secondary metabolite biosynthesis pathways, including Biosynthesis of secondary metabolites (ko01110), Flavonoid biosynthesis (ko00941), and Anthocyanin biosynthesis (ko00942) (Fig 6B). Notably, key enzyme genes in the flavonoid pathway, encoding phenylalanine ammonia-lyase (*PAL*), chalcone isomerase (*CHI*), chalcone synthase (*CHS*) and anthocyanin synthase (*ANS*), were significantly upregulated, corroborating the explosive accumulation of cyanidin derivatives (FC > 139) observed in metabolomic data.

In conclusion, WLc treatment induces the expression of genes in the core phenylpropanoid pathway (from *PAL* to *CHS*, and *ANS*), upregulates key anthocyanin biosynthesis genes, and ultimately drives preferential accumulation of cyanidin-derived pigments in *L. leichtlinii* subsp. *maximowiczii* bulbs. This integrated multi-omics analysis provides comprehensive insights into light-regulated bulb coloration mechanisms, establishing a molecular foundation for targeted improvement of ornamental traits in lilies.

### 3.7  Temporally coordinated enzymatic cascade drives light-induced anthocyanin assembly

To explore the molecular mechanisms of anthocyanin biosynthesis in *L. leichtlinii* subsp. *maximowiczii* bulbs under WLc storage, we profiled the expression of genes encoding key pigment metabolic enzymes (Fig 7). Among 53 flavonoid biosynthesis-related genes detected, 39 were significantly upregulated under WL6, spanning phenylpropanoid metabolism including *PAL, C4H* (Cinnamate 4-hydroxylase), *CHS, CHI*, flavonoid modification genes *F3H* (Flavanone 3-hydroxylase), *F3'H* (Flavonoid 3'-hydroxylase), *FLS* (Flavonol synthase), and anthocyanin end-product synthesis genes *DFR* (Dihydroflavonol 4-reductase), *ANS, 3GT, BZ1* (Bronze 1). qRT-PCR validation confirmed that 17 of 25 candidate genes exhibited

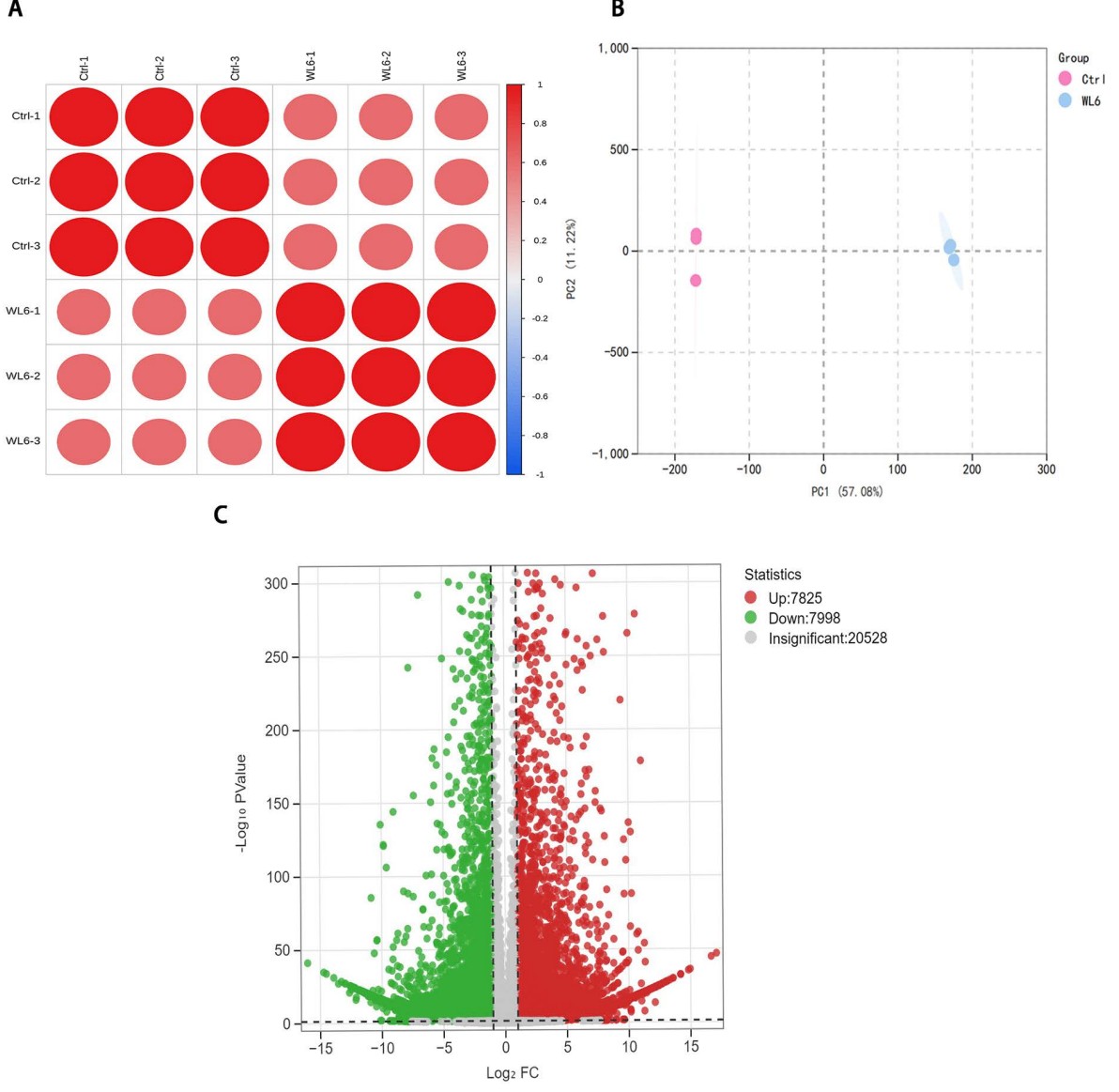

**Fig 5. Overview of transcriptome data quality and differential expression analysis. (A)** Inter-sample correlation heatmap (Pearson *r* > 0.99). **(B)** Principal component analysis (PCA) of Ctrl vs. WL6 groups. **(C)** Volcano plot of differentially expressed genes (DEGs) (Fold Change > 2.0, *p* < 0.05). Ctrl: day 0; WL6: 6-day white light.

expression trends highly consistent with RNA-seq data (Fig 8), with *CHS2* (Cluster-27448.10), *CHSC-2* (Cluster-2480.9), *DFR* (Cluster-2527.10), *FLS-2* (Cluster-1772.0), and *ANS* (Cluster-22832.0) displaying significant positive correlations with anthocyanin accumulation.

Enzymatic activity analysis revealed distinct temporal dynamics: *DFR* and *FLS-2* mediate the reduction of dihydroflavonols to leucoanthocyanidins and their oxidation to flavonols, respectively. After 9 days of WLc treatment, their FPKM values surged to 1,425.78 and 974.83, representing ~1,425-fold increase over controls. *F3H* reached 8.59-fold higher than day 0 by day 3, driving the conversion of naringenin to dihydrokaempferol and providing precursors for anthocyanin

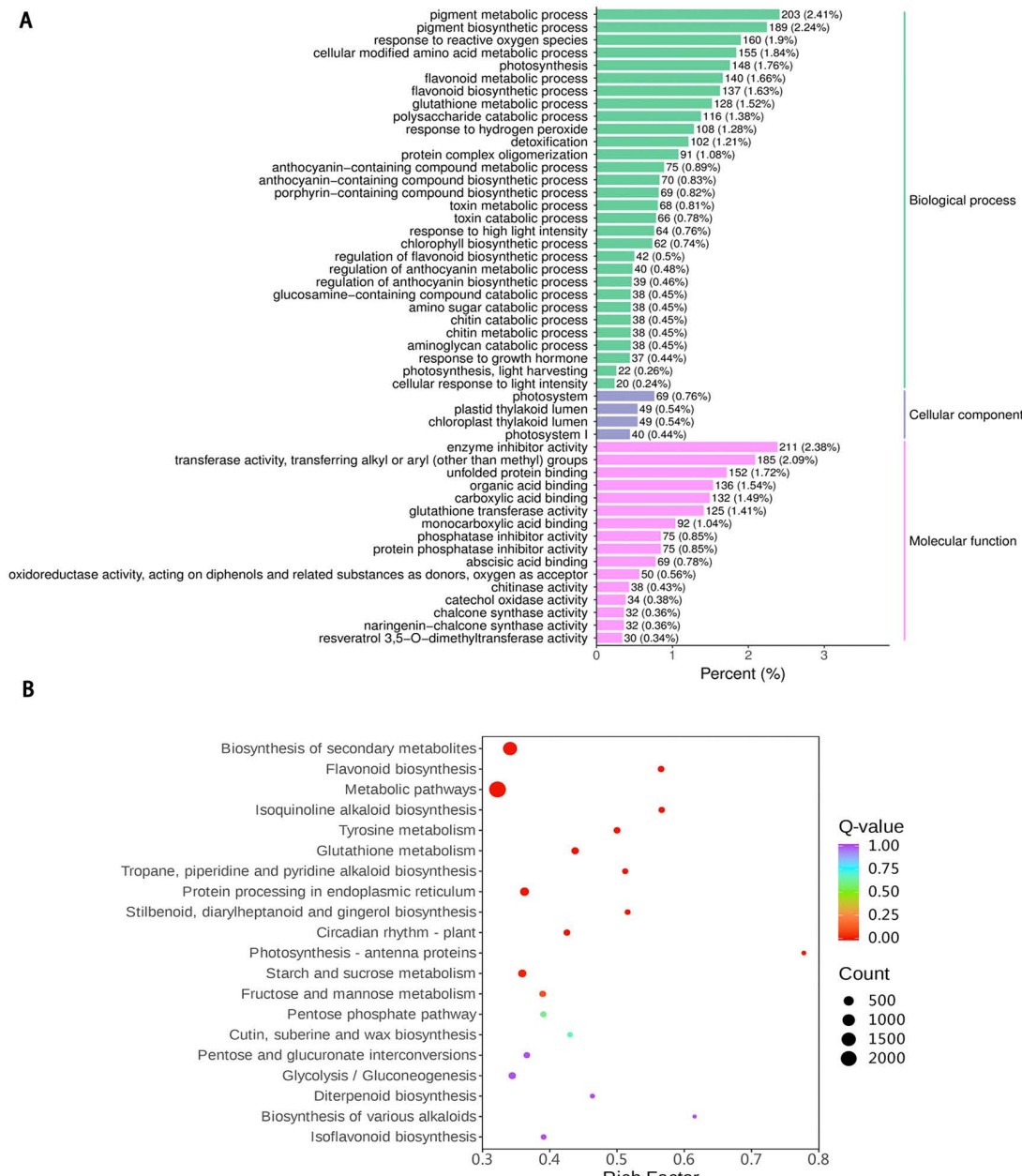

**Fig 6. GO functional classification and KEGG pathway enrichment analysis of differentially expressed genes. (A)** Gene Ontology (GO) classification of DEGs (top 50 terms). **(B)** KEGG pathway enrichment.

synthesis (Fig 8L). *ANS* (anthocyanidin synthase) exhibited linear expression increases, peaking at 191.25 FPKM on day 9, directly catalyzing the oxidation of leucoanthocyanidins to anthocyanidin aglycones (Fig 8R). UDP-glucose: flavonoid 3-glucosyltransferase (*3GT*) was significantly induced by day 3, with sustained high expression (days 6–9) promoting anthocyanidin glycosylation to enhance pigment stability (Fig 8X).

Time-course expression revealed that upstream phenylpropanoid genes (*CHS, CHI*) and branchpoint-controlling genes (*F3'H, ANR-1*) peaked at day 3, while downstream genes (*DFR, ANS*) showed gradual induction. This coordinated

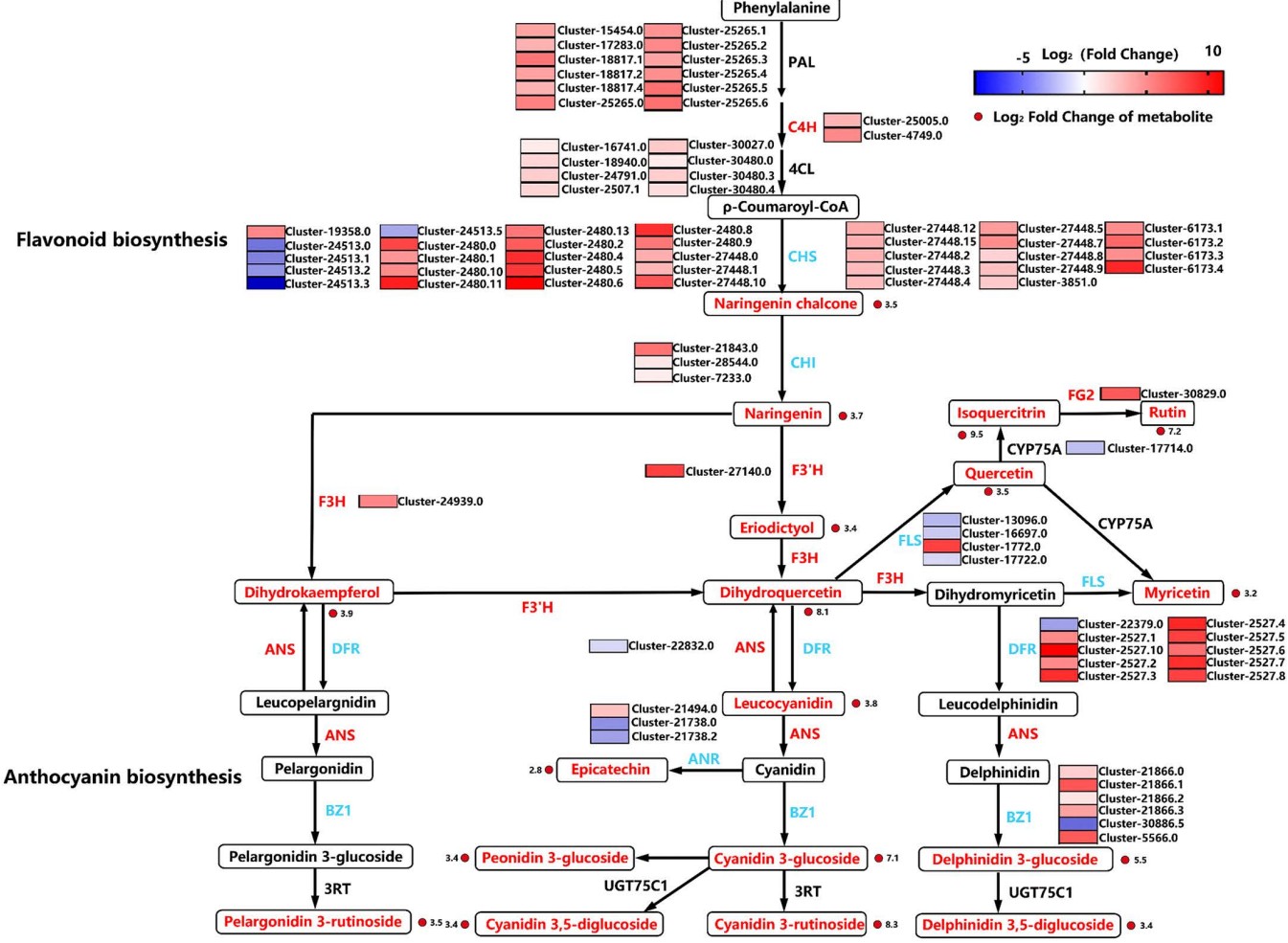

**Fig 7. Transcriptional reprogramming of anthocyanin-flavonoid pathways in *Lilium leichtlinii* subsp. *maximowiczii* bulbs under white light storage.** Gene expression changes are represented by a blue-white-red heatmap showing log₂(Fold Change) from −5 to 10, with significant transcripts meeting Fold Change > 2.0, $p < 0.05$. Gene families are marked with red borders if all isoforms are upregulated and blue borders for families containing both up- and down-regulated isoforms. Metabolites confirmed by metabolomics are labeled in red text, while undetected compounds appear in black. Solid red circles denote metabolites with significant accumulation changes ($|\log_2 FC| > 1$, $p < 0.05$).

enzymatic cascade preceded and facilitated the rapid synchronized anthocyanin accumulation in metabolomic data, indicating that WLc-induced bulb coloration involves temporally coordinated activation of enzymatic nodes. These findings elucidate the molecular basis of light-mediated anthocyanin assembly, providing targets for precision breeding of ornamental lilies with enhanced color stability.

### 3.8 Evolutionary conservation of catalytic architecture in anthocyanin biosynthetic enzymes

Phylogenetic analysis of key anthocyanin biosynthetic enzymes was conducted using de novo trees constructed in MEGA11 (Neighbor-Joining method, bootstrap = 1000 replicates), revealing evolutionary conservation patterns in *L. leichtlinii* subsp. *maximowiczii*. The LlF3H protein clustered within a well-supported clade with its ortholog in *Tulipa fasteriana* (Liliaceae), whereas longer evolutionary distances were observed compared to eudicot orthologs such as *Arabidopsis thaliana* (Brassicaceae) and *Apocynum venetum* (Apocynaceae) (S5 Fig A). Similarly, LlANS protein formed a

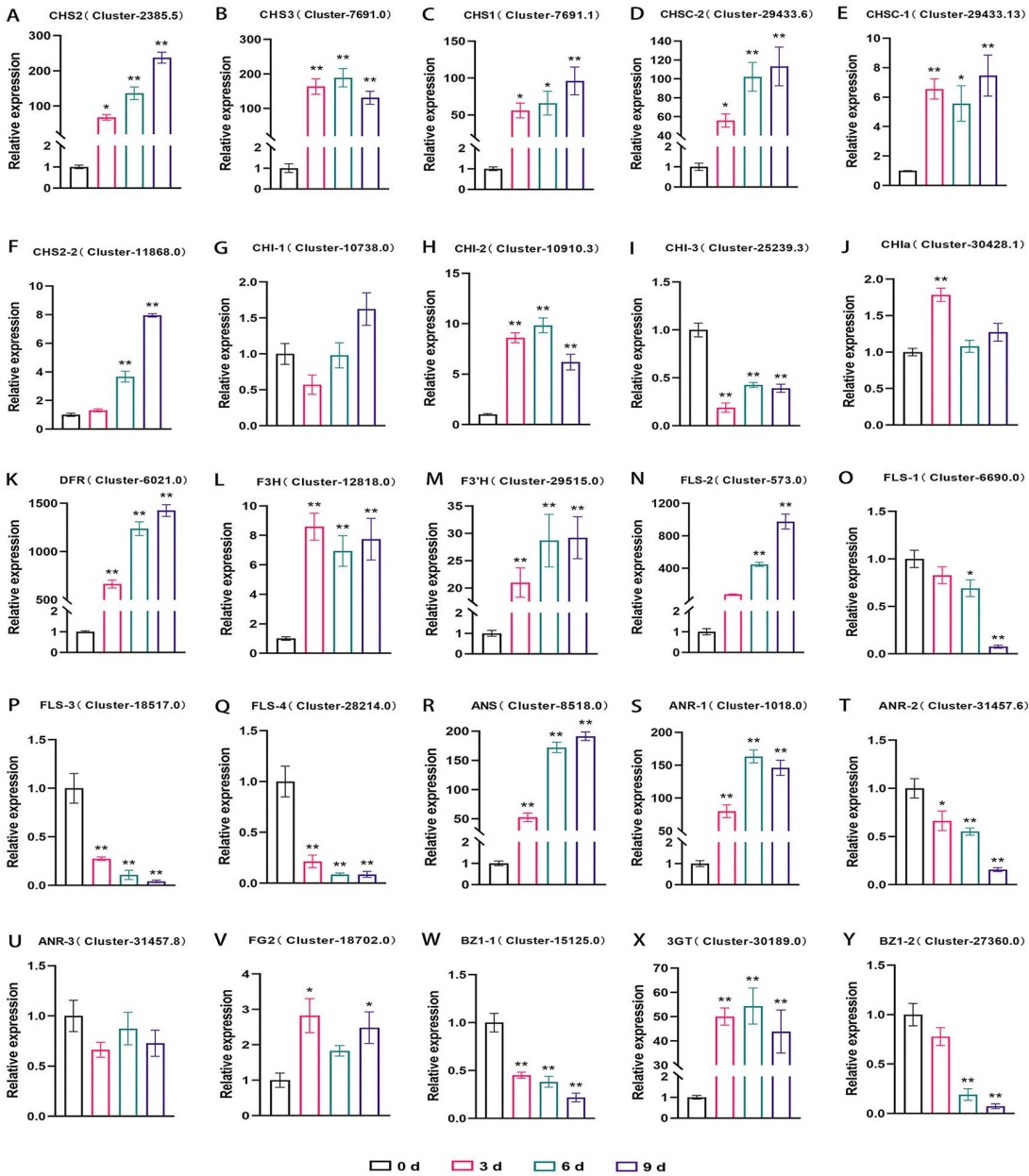

**Fig 8. qRT-PCR validation of anthocyanin biosynthetic gene expression dynamics under white-light treatment.** Time-course analysis of key enzyme genes at 0, 3, 6, and 9 days. Data were normalized to *Actin* and *18S* reference genes and presented as mean ± SD (n = 3). Asterisks indicate significant differences compared to 0-day controls (*, $p < 0.05$; **, $p < 0.01$) based on two-way ANOVA with Tukey's test.

monophyletic group with *Tulipa fasteriana* 'Emperor' (Liliaceae), *Lycoris longituba* (Amaryllidaceae), and *Strelitzia reginae* (Strelitziaceae), but showed significant divergence from *A. thaliana* and *Platycodon grandiflorus* (Campanulaceae) (S5 Fig B). Ll3GT protein clustered within a liliaceous clade (*Erythronium*, *Tulipa*), sharing higher sequence identity with *Lilium hybrid* 3GT (>97%) but maximal divergence from *Citrus sinensis* (Rutaceae) and *Camellia sinensis* (Theaceae) (S5 Fig C). These results confirm that anthocyanin enzyme evolution aligns with angiosperm phylogeny, with functional conservation within Liliaceae.

Multiple sequence alignment (DNAMAN) of amino acid sequences further demonstrated conserved structural features across anthocyanin biosynthetic enzymes in *L. leichtlinii* subsp. *maximowiczii*. The LlF3H protein showed >92% residue conservation in the 2-OG oxygenase domain compared to orthologous proteins from four species, with 95.9% domain identity to TmF3H (S5 Fig D), indicating functional conservation in flavonoid metabolism. LlANS protein showed complete overlap of its 2-OG oxygenase domain with three structural homologs. Domain identity with TmANS reached 94.9% (S5 Fig E), suggesting evolutionary stability in anthocyanidin oxidation. The Ll3GT protein contained the canonical magnesium transporter (MGT) motif in its sugar-binding domain, demonstrating 97.6% domain identity with Lh3GT (S5 Fig F), which supports conserved glycosyltransferase activity. These protein alignment results demonstrate that key anthocyanin bio-synthetic enzymes in *L. leichtlinii* subsp. *maximowiczii* possess evolutionarily conserved catalytic cores, providing a robust structural basis for functional characterization and targeted engineering of ornamental traits.

## 4. Discussion

### 4.1 Light quality-driven phenotypic and metabolic remodeling

Our results demonstrate that light quality differentially modulates color evolution and secondary metabolism in *L. leichtlinii* subsp. *maximowiczii* bulbs. WL and BL treatments induced violet-red pigmentation, likely mediated through the activation of cryptochrome (CRY) or phytochrome (phy) photoreceptor activation, which upregulates phenylpropanoid genes and triggers anthocyanin deposition in vacuoles [19,20]. In contrast, RL led to a yellowish – brown pigmentation, with a b* value increase of 14.4. This pigmentation might stem from the activation of carotenoid or proanthocyanidin biosynthesis pathways [21]. RL may also indirectly regulate the flavonoid pathway via the phyB-mediated jasmonic acid (JA) signaling [22]. Dark controls showed minimal metabolic changes, confirming light as an essential trigger for pigmentation [23]. Nota-bly, BL rapidly elevated total phenolics during early storage, indicating differential regulation of phenylpropanoid branches. GL specifically enhanced anthocyanins (32-fold by day 3) without affecting total phenolics, suggesting phyC potentially mediates metabolic partitioning [24]. These mechanisms provide a molecular framework for developing spectrum-specific preservation technologies to optimize ornamental bulb quality.

### 4.2 Temperature and packaging constraints on obligate light-driven pigmentation

Temperature and packaging conditions modulated light-dependent pigmentation in *L. leichtlinii* subsp. *maximowiczii* bulbs. Under WLc, bulbs stored at both 4 °C and 25 °C developed violet-red pigmentation, while 37 °C induced yellowish-brown discoloration likely due to thermoinactivation of anthocyanin biosynthetic enzymes [25]. Under dark treatment, no violet-red pigmentation occurred regardless of temperature, confirming that temperature modulates but does not initiate pigmentation [26].

Comparative analysis revealed WL-induced violet-red pigmentation in NP bulbs, accompanied by elevated anthocy-anin, flavonoid, and total phenolic content. Under dark treatment, vacuum packaging effectively inhibited the occurrence of violet-red pigmentation in lily bulbs and significantly suppressed the accumulation of secondary metabolites, likely through oxygen depletion inhibiting enzymatic oxidation reactions, while dark storage reduced phenolic synthesis and enzyme activation [27]. The combined effects of VP and Dc compromised anthocyanin biosynthesis, demonstrating that oxygen availability and light signaling jointly regulate pigment metabolism in lily bulbs during storage.

Comprehensive anthocyanin analyses confirmed that biosynthesis is strictly light-dependent in *Lilium leichtlinii* subsp. *maximowiczii* bulbs, with light serving as the primary inducer of violet-red pigmentation. Temperature and packaging act as secondary modulators of this process, as illustrated in the integrated model (Fig 9).

### 4.3 Coordinated transcriptional-metabolic remodeling drives light-induced anthocyanogenesis

Integrated metabolomic-transcriptomic analysis uncovered a hierarchical regulatory network underlying WLc-induced anthocyanin synthesis in *L. leichtlinii* subsp. *maximowiczii* bulbs. Metabolomic profiling confirmed selective accumulation

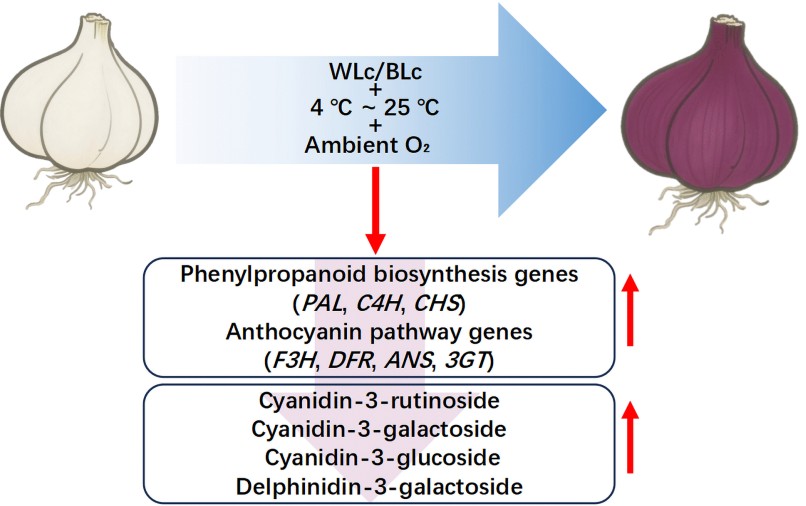

**Fig 9. Under combined white and blue light (WLc/BLc), within 4−25 °C under ambient O$_2$, the bulb epidermis of *Lilium leichtlinii* subsp. *maximowiczii* develops violet-red pigmentation.** This process involves upregulation of phenylpropanoid biosynthesis genes (*PAL*, *C4H*, *CHS*) and anthocyanin pathway genes (*F3H*, *DFR*, *ANS*, *3GT*), resulting in accumulation of cyanidin derivatives, specifically cyanidin-3-rutinoside, cyanidin-3-galactoside and cyanidin-3-glucoside, along with delphinidin-3-galactoside and associated phenylpropanoid metabolites.

of cyanidin derivatives (e.g., cyanidin-3-rutinoside, FC = 316.53), stabilized through 3GT-mediated glycosylation [28]. KEGG enrichment indicated coordinated activation of phenylpropanoid and flavonoid pathways (RF = 0.38), establishing the metabolic basis for anthocyanin surge [29]. Transcriptomic analysis identified 20,777 DEGs, with core phenylpropanoid genes (*PAL*, *CHS*, *ANS*) exhibiting significant upregulation and expression dynamics directly correlating with metabolite accumulation patterns, supporting their direct roles in anthocyanin production [30]. Notably, four enzyme-coding genes (*CHS, DFR, FLS, ANS*) previously identified as key determinants of rose petal coloration in transcriptomic studies [31], showed delayed *FLS-2* expression after day 3, suggesting substrate competition mechanisms that balance flavonol and anthocyanin synthesis, reflecting a physiological trade-off between pigmentation and defense [32].

## 4.4 Structural and temporal architecture of conserved catalytic domains informs metabolic engineering

Time-series expression analysis uncovered a light-driven enzymatic cascade: upstream genes (*CHS*, *CHI*) peaked at day 3 to initiate flavonoid skeleton assembly, while downstream genes (*DFR*, *ANS*) showed gradual induction for precursor conversion. The 1,400-fold upregulation of *DFR* (1,425.78 FPKM), potentially regulated by MYB transcription factors, supplied leucoanthocyanidins to *ANS* (191.25 FPKM). The linear accumulation of *ANS* transcripts directly correlated with anthocyanin deposition [33–35]. Phylogenetic and multiple sequence alignment analyses revealed that the key enzymes LIF3H and LIANS maintain high cross-species conservation in the catalytic 2-OG oxygenase domain core. However, within the substrate recognition region, the MGT motif of LI3GT shows adaptive divergence [36]. This "conserved core, variable periphery" pattern explains metabolic pathway conservation within *Liliaceae* (e.g., *Tulipa* clustering) and regulatory divergence in distant species (e.g., *Arabidopsis*), providing structural insights for cross-species metabolic engineering. *LIANS* clustering with *Strelitzia* suggests convergent evolution of monocot oxidases, though substrate preference (e.g., cyanidin vs. delphinidin) may be modulated by variable regions. Future studies could employ CRISPR editing of ANS substrate-binding domains or optimized light combinations (e.g., 450 nm blue + 380 nm UV) to enhance pigment yield. These findings establish a structure-guided framework for targeted improvement of lily color traits and high-value metabolite production, advancing precision breeding strategies in ornamental horticulture.

## 5. Conclusion

This study deciphers the molecular mechanisms governing light spectrum-regulated coloration and secondary metabolism in *L. leichtlinii* subsp. *maximowiczii* bulbs. White (WL) and blue light (BL) cooperatively activate upstream phenylpropanoid genes (*PAL*, *C4H*, *CHS*) and downstream anthocyanin genes (*DFR*, *ANS*, *3GT*), driving rapid accumulation of cyanidin derivatives (e.g., cyanidin-3-rutinoside, 12-fold higher than dark controls) and resulting in violet-red pigmentation (ΔE = 55.2 and 52.1 at day 12). Red light (RL) preferentially induced flavonoid accumulation, causing yellowish-brown discoloration (b* = +14.4 units), while GL specifically enhanced anthocyanins without affecting total phenolic content. Dc suppressed metabolic activity, and high temperature (37 °C) or vacuum packaging partially inhibited pigmentation. Integrated omics revealed WL-mediated metabolic rewiring via light-induced transcriptional cascades, culminating in metabolic reprogramming. Transcriptomic profiling confirmed significant enrichment in flavonoid biosynthesis (KEGG, $p < 0.01$), with co-expression of *DFR*, *F3H*, *ANS*, *BZ1* directly correlating with metabolite dynamics. Phylogenetic analyses demonstrated >94.9% conservation of catalytic domains (2-OG oxygenase domain, MGT motif) in anthocyanin enzymes (*LIF3H*, *LIANS*, *LI3GT*) across Liliaceae species. This work establishes a spectral regulation network model that integrates light quality, temperature, and packaging constraints, providing a mechanistic basis for edible lily color improvement and spectrum-controlled postharvest technologies. The findings advance precision breeding strategies by elucidating how environmental factors interact to control metabolic flux and pigment stability, offering actionable targets for optimizing ornamental traits and metabolite production in horticultural crops.

## Supporting information

**S1 Fig. Colorimetric dynamics of *Lilium leichtlinii* subsp. *maximowiczii* bulbs during storage under light quality regulation.** (A) Lightness (L*) dynamics. (B) Red-green coordinate (a*) dynamics. (C) Yellow-blue coordinate (b*) dynamics. (D) Total color difference (ΔE) values. Capital letters (A–F) indicate significant differences across storage days (0–12 d) under identical light; lowercase letters (a–e) indicate significant differences among light treatments (WLc, BLc, GLc, RLc, Dc) at identical storage time ($p < 0.05$ by two-way ANOVA with Tukey's test). Data presented as mean ± SD (n = 5). WLc: white light; BLc: blue light; RLc: red light; GLc: green light; Dc: dark control.
(TIF)

**S2 Fig. Time-course dynamics of primary metabolites in *Lilium leichtlinii* subsp. *maximowiczii* bulbs under spectral treatments.** (A) Total soluble sugars (mg/g). (B) Sucrose (mg/g). (C) Starch (mg/g). (D) Total amino acids (mg/g). (E) Vitamin C (μg/mg protein). Asterisks indicate significant differences between light treatments at each time point (*, $p < 0.05$; **, $p < 0.01$; two-way ANOVA with Tukey's test). Data points represent as mean ± SD (n = 3). WLc: white light; BLc: blue light; GLc: green light; RLc: red light; Dc: dark control. X-axis: Days of treatment (0, 1, 3, 6, 9, 12).
(TIF)

**S3 Fig. Temperature-modulated colorimetric dynamics of *Lilium leichtlinii* subsp. *maximowiczii* bulbs under spectral treatments.** (A, B) Lightness (L*) dynamics. (C, D) Red-green coordinate (a*) dynamics. (E, F) Yellow-blue coordinate (b*) dynamics. (G, H) Total color difference (ΔE) values. Capital letters (A–F) indicate significant differences across storage days (0–12 d) within the same temperature; lowercase letters (a–c) indicate significant differences among temperatures (4 °C, 25 °C, 37 °C) at identical storage time ($p < 0.05$ by two-way ANOVA with Tukey's test). Data presented as mean ± SD (n = 5). WLc: white light; Dc: dark control.
(TIF)

**S4 Fig. Packaging-modulated colorimetric dynamics in *Lilium leichtlinii* subsp. *maximowiczii* bulbs under spectral treatments.** (A, B) Lightness (L*) dynamics. (C, D) Red-green coordinate (a*) dynamics. (E, F) Yellow-blue coordinate (b*) dynamics. (G, H) Total color difference (ΔE) values. Capital letters (A-F): Differences across storage days (0–12 d)

within same packaging; lowercase letters (a-b): Differences among packaging types (VP, NP) at identical storage time ($p < 0.05$ by two-way ANOVA with Tukey's test). Data presented as mean±SD (n=5). Packaging: VP (vacuum packaging); NP (normal packaging). WLc: white light; Dc: dark control.
(TIF)

**S5 Fig. Phylogenetic and structural conservation analysis of anthocyanin biosynthetic enzymes in *Lilium leichtlinii* subsp. *maximowiczii*.** (A) Neighbor-Joining phylogenetic tree of F3H orthologs. Number: bootstrap values. (B) Phylogenetic tree of ANS orthologs. Number: bootstrap values. (C) Phylogenetic tree of 3GT orthologs. Number: bootstrap values. (D) Multiple sequence alignment of F3H proteins. Black bar: Conserved 2-OG oxygenase domain. (E) Multiple sequence alignment of ANS proteins. Black bar: Catalytic 2-OG oxygenase domain. (F) Multiple sequence alignment of 3GT proteins. Black bar: MGT motif in glycosyltransferase domain. LlF3H: *Lilium leichtlinii* subsp. *maximowiczii*; AtF3H: *Arabidopsis thaliana* (OAP00169.1); MdF3H: *Malus domestica* (AAX89397.1); RcF3H: *Rubus coreanus* (ABW74548.1); TfF3H: *Tulipa fosteriana* (AGJ50588.1); LlANS: *Lilium leichtlinii* subsp. *maximowiczii*; AtANS: *Arabidopsis thaliana* (OAO93537.1); SrANS: *Strelitzia reginae* (AGC73738.1); TfANS: *Tulipa fosteriana* (AGJ50591.1); Ll3GT: *Lilium leichtlinii* subsp. *maximowiczii*; At3GT: *Arabidopsis thaliana* (OAP13723.1); Eu3GT: *Erythronium umbilicatum* (UWK01865.1); Lh3GT: *Lilium hybrid* (ARR28805.1); Tf3GT: *Tulipa fosteriana* (AHY20031.1).
(TIF)

**S1 Table. Parameters and statistical tables for transcriptome validation and metabolomics analysis.** 1. Primers for verification of L-bulb transcriptome by qRT-PCR; 2. The qRT-PCR reaction system; 3. for qRT-PCR reaction procedures; 4. Second-class classification of the metabolites; 5. Statistical Table of KEGG Pathways (log2FC > 4.79); 6. Statistical table of the assembly results; 7. annotation statistics.
(DOCX)

## Author contributions

**Conceptualization:** xinhai Yu, tianliang Wang.

**Data curation:** shuai Li, tianliang Wang.

**Formal analysis:** jinxing xu.

**Investigation:** chen Wang, qi wu, yanli Ma, yuanhang Zhou.

**Methodology:** xinhai Yu.

**Project administration:** tianliang Wang.

**Software:** chen Wang, bingyan liu, xinzhu Dai.

**Supervision:** bingyan liu, qi wu, xinzhu Dai, yuanhang Zhou.

**Validation:** jinxing xu, shuai Li, yanli Ma.

**Writing – original draft:** jinxing xu, shuai Li.

**Writing – review & editing:** xinhai Yu, tianliang Wang.

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
