## [Decision Letter · Decision Letter 0]

2 Mar 2026

PONE-D-25-49782Light Quality-Regulated Anthocyanin Biosynthesis in Lilium leichtlinii var.  maximowiczii Bulbs: A Multi-Omics PerspectivePLOS One

Dear Dr. xu,

Thank you for submitting your manuscript to PLOS ONE. After careful consideration, we feel that it has merit but does not fully meet PLOS ONE’s publication criteria as it currently stands. Therefore, we invite you to submit a revised version of the manuscript that addresses the points raised during the review process.

If applicable, we recommend that you deposit your laboratory protocols in protocols.io to enhance the reproducibility of your results. Protocols.io assigns your protocol its own identifier (DOI) so that it can be cited independently in the future. For instructions see: https://journals.plos.org/plosone/s/submission-guidelines#loc-laboratory-protocols. Additionally, PLOS ONE offers an option for publishing peer-reviewed Lab Protocol articles, which describe protocols hosted on protocols.io. Read more information on sharing protocols at . Additionally, PLOS ONE offers an option for publishing peer-reviewed Lab Protocol articles, which describe protocols hosted on protocols.io. Read more information on sharing protocols at https://plos.org/protocols?utm_medium=editorial-email&utm_source=authorletters&utm_campaign=protocols..

We look forward to receiving your revised manuscript.

Kind regards,

Branislav T. Šiler, Ph.D.

Academic Editor

PLOS One

Journal Requirements:

“This study was supported by Science and Technology Development Program of Jilin Province (2024JB427L35).”

5. Please note that funding information should not appear in any section or other areas of your manuscript. We will only publish funding information present in the Funding Statement section of the online submission form. Please remove any funding-related text from the manuscript.

6. In the online submission form, you indicated that your data is available only on request from a third party. Please note that your Data Availability Statement is currently missing [the name of the third party contact or institution / contact details for the third party, such as an email address or a link to where data requests can be made]. Please update your statement with the missing information.

7. Please amend the manuscript submission data (via Edit Submission) to include author “Shuai Li , Cheng Wang, Bingyan Liu , Qi Wu, Yanli Ma, Xinzhu Dai, Yuanhang Zhou, Xinhai Yu, Tianliang Wang”.

8. We notice that your supplementary figures are uploaded with the file type 'Figure'. Please amend the file type to 'Supporting Information'. Please ensure that each Supporting Information file has a legend listed in the manuscript after the references list.

Additional Editor Comments:

Both reviewers noticed that the methodology requires more detailed description in several specific points.

Please mind that *Lilium leichtlinii* subsp.  subsp. *maximowiczii* (Regel) J.Compton is a subspecies, not a variety (see e.g., https://powo.science.kew.org/taxon/urn:lsid:ipni.org:names:77215715-1).(Regel) J.Compton is a subspecies, not a variety (see e.g., https://powo.science.kew.org/taxon/urn:lsid:ipni.org:names:77215715-1).

"*Lilium* spp." cannot contain 155 species, but the genus  spp." cannot contain 155 species, but the genus *Lilium* can contain them. "Lilium" in italics (e.g. L48). "unicolor" in italics (L63). can contain them. "Lilium" in italics (e.g. L48). "unicolor" in italics (L63).

I suggest dividing Figure 5 into several, more easily readable figures, since the font size in D and E is barely visible.

The authors are strongly encouraged to have the manuscript proofread by a native English speaker or a professional editing agency to polish the text and improve general readability.

Reviewer's Responses to Questions

**Comments to the Author**

1. Is the manuscript technically sound, and do the data support the conclusions?

Reviewer #1: Yes

Reviewer #2: Yes

2. Has the statistical analysis been performed appropriately and rigorously? 

Reviewer #1: Yes

Reviewer #2: Yes

3. Have the authors made all data underlying the findings in their manuscript fully available?

Reviewer #1: Yes

Reviewer #2: Yes

4. Is the manuscript presented in an intelligible fashion and written in standard English?

Reviewer #1: No

Reviewer #2: Yes

5. Review Comments to the Author

Reviewer #1: 1. The manuscript reports large fold changes (e.g., cyanidin-3-rutinoside FC = 316.53, DFR FPKM = 1425.78) (line no. 402) that appear unusually high; ensure normalization and log-transformation are appropriately applied.

2. De-italicize Bulb to Bulb wherever necessary.

3. The Introduction section contains several sentences written in the future tense (e.g., “This study will elucidate…”). Please revise these to present tense to accurately reflect completed research and maintain consistency with standard scientific writing conventions.

4. Section 2.1.2: The methodology requires more detailed description—please include specific temperature control conditions (e.g., type of incubator, monitoring frequency, humidity). Additionally, spell out the abbreviations Dc and WLc in full when they first appear, and consider removing or minimizing abbreviations throughout this section for improved readability and clarity.

5. Kindly clarify whether the samples used for RNA seq, q-RT PCR and metabolomics were taken from same batch or different batch and also mention clearly the stages of sample preparation for the same.

6. Clearly mention the number of replicates taken to perform transcriptomics, metabolomics and qRT-PCR in the material and methods section.

7. The font size of figures should be improved to enhance readability and clarity and include bootstrap values in the phylogenetic tree.

8. Line number 310 word “Ctrl” has not mentioned anywhere in the material and method section.

Reviewer #2: The paper provides a comprehensive investigation of the regulation of anthocyanins in lilies through omics approaches and contains a substantial amount of information. However, the experimental design with regard to temperature requires refinement. The paper is acceptable overall, and minor revisions are recommended for acceptance.

1.References to research on Lilium leichtlinii var. maximowiczii should be added between lines 68 and 73. For instance, it would be beneficial to readers if its medicinal value were clarified.

2.The word 'bulbs' does not require italics on line 93.

3.Line98&Line107. Please explain why a temperature of 10°C was selected for the light treatment and why 4°C was chosen as one of the three temperatures. What considerations underlie these decisions?

4.Line 140: Freeze-dried lilies should be ground into powder while they are still under liquid nitrogen conditions.

5.How should one explain the fact that metabolic analysis only uses samples from the six-day treatment period? And why is this the case?

6.Line 177: The thermo instruments should be labelled with their model and the country of manufacture.

7.The cDNA reverse transcription step should be included on line 188.

8.The use of the internal reference gene should be supplemented on line 196.

9.What is the explanation for the build-up of anthocyanins at higher temperatures, yet the observation that the effect at 37°C is less marked than at 25°C?

10.In conclusion, line 545 is best applied to edible lilies.

6. PLOS authors have the option to publish the peer review history of their article (what does this mean?). If published, this will include your full peer review and any attached files.). If published, this will include your full peer review and any attached files.

If you choose “no”, your identity will remain anonymous but your review may still be made public

.

Reviewer #1: **Yes:**Bhavya BhargavaBhavya Bhargava

Reviewer #2: **Yes:**Zhimin LinZhimin Lin

---

## [Author Response · Author response to Decision Letter 1]

2 Apr 2026

Reviewer: 1

1. The manuscript reports large fold changes (e.g., cyanidin-3-rutinoside FC = 316.53, DFR FPKM = 1425.78) (line no. 402) that appear unusually high; ensure normalization and log-transformation are appropriately applied.

Response: We thank the reviewer for raising this concern. Regarding the large fold changes reported in the manuscript, we have re-examined the procedures for data normalization and log transformation. The verification results confirm that these high fold changes truly reflect significant differences in the expression levels of the corresponding metabolites and genes between the treatment and control groups, with consistent measurements across biological replicates.

2.De-italicize Bulb to Bulb wherever necessary.

Response: Thank you for your careful review. As suggested, we have removed the italics from “Bulb” throughout the manuscript to comply with standard formatting guidelines (line 100).

3.The Introduction section contains several sentences written in the future tense (e.g., “This study will elucidate…”). Please revise these to present tense to accurately reflect completed research and maintain consistency with standard scientific writing conventions.

Response: We thank the reviewer for the careful reading. As suggested, we have changed the tense in the Introduction section from future to present tense to better conform to standard academic writing conventions. For example, in the revised manuscript, "This study will systematically elucidate" has been changed to "This study systematically elucidates" (Line 84). All changes have been highlighted in the revised manuscript.

4.Section 2.1.2: The methodology requires more detailed description—please include specific temperature control conditions (e.g., type of incubator, monitoring frequency, humidity). Additionally, spell out the abbreviations Dc and WLc in full when they first appear, and consider removing or minimizing abbreviations throughout this section for improved readability and clarity.

Response: We thank the reviewer for their valuable suggestions. Accordingly, we have supplemented Section 2.1.2 with specific details on the temperature control conditions, including the equipment used and the humidity control range. In addition, to enhance readability and clarity, we have spelled out the abbreviations “Dc” and “WLc” in full upon their first appearance in the text (lines 106–108).

5.Kindly clarify whether the samples used for RNA seq, q-RT PCR and metabolomics were taken from same batch or different batch and also mention clearly the stages of sample preparation for the same.

Response:We thank the reviewer for raising this critical point regarding sample consistency. To clarify, the samples used for RNA sequencing, qRT-PCR, and metabolomics analysis were all derived from the same batch of experiments. Specifically, all samples were subjected to identical treatment conditions to ensure comparability across the different omics datasets. This information has now been added in Section 2.6.

6.Clearly mention the number of replicates taken to perform transcriptomics, metabolomics and qRT-PCR in the material and methods section.

Response: Thank you for pointing this out. In the transcriptomics, metabolomics, and qRT‑PCR experiments, three technical replicates were set for each sample to ensure reproducibility and reliability. The corresponding replicate numbers have now been added to the Materials and Methods section.

7.The font size of figures should be improved to enhance readability and clarity and include bootstrap values in the phylogenetic tree.

Response: Thank you for your valuable comments. We have improved the font size of the figures to enhance readability and clarity, and we have also included bootstrap values in the phylogenetic tree.

8. Line number 310 word “Ctrl” has not mentioned anywhere in the material and method section.

Response: Thank you for your insightful comment. In the manuscript, “Ctrl” denotes the control group used in the experiments. To address this concern, we have now added a clarification in the Materials and Methods section (line 164) to ensure that readers can clearly understand this abbreviation. We appreciate your valuable suggestion, which has helped improve the clarity and rigor of our work.

Reviewer: 2

Comments to the Author

The paper provides a comprehensive investigation of the regulation of anthocyanins in lilies through omics approaches and contains a substantial amount of information. However, the experimental design with regard to temperature requires refinement. The paper is acceptable overall, and minor revisions are recommended for acceptance.

Major Revisions

1. References to research on Lilium leichtlinii var. maximowiczii should be added between lines 68 and 73. For instance, it would be beneficial to readers if its medicinal value were clarified.

Response: Thank you very much for your valuable suggestion. We have added relevant references regarding Lilium leichtlinii var. maximowiczii between lines 78 and 79 to highlight its potential medicinal value and enhance readers' understanding of the significance of this species.

1. Wang, Y., et al., Traditional uses, nutritional properties, phytochemical metabolites, pharmacological properties, and potential applications of Lilium spp.: a systematic review. Frontiers in Pharmacology, 2025. 16: p. 1713957.

2. The word 'bulbs' does not require italics on line 93.

Response: Thank you for your careful review. As suggested, we have removed the italics from “Bulb” throughout the manuscript to comply with standard formatting guidelines （ line 100 ）.

3. Line98&Line107. Please explain why a temperature of 10°C was selected for the light treatment and why 4°C was chosen as one of the three temperatures. What considerations underlie these decisions?

Response: We thank the reviewer for raising this question. The temperature of 10°C was selected because it represents a common critical temperature during plant growth or postharvest storage. For crops such as edible lily, this temperature effectively extends the shelf life and delays senescence without causing significant chilling injury, thereby facilitating the observation of the independent effects of light on color regulation. This approach helps ensure that the observed physiological changes can be primarily attributed to the light treatment itself, rather than to damage caused by low temperature.

The temperature of 4°C was chosen as it represents a standard condition for low-temperature stress or chilling injury. By including this temperature, we can clearly characterize the physiological responses of edible lily under extreme low-temperature stress, such as whether anthocyanins accumulate or degrade abnormally in response to such stress.

4. Line 140: Freeze-dried lilies should be ground into powder while they are still under liquid nitrogen conditions.

Response: Thank you for your valuable comment. We confirm that the freeze-dried lily samples were indeed ground into powder under liquid nitrogen conditions. This information was inadvertently omitted in the original manuscript and has now been added at line 147 in the revised version. We appreciate your careful review and the opportunity to improve our manuscript.

5. How should one explain the fact that metabolic analysis only uses samples from the six-day treatment period? And why is this the case?

Response: Thank you for your insightful question regarding the selection of samples from the 6-day treatment period for metabolomic analysis. Our decision was based on experimental observations and quality assessments. Specifically, we observed a distinct color change in the lily bulbs on day 6, and the overall quality at this time point was superior to that of the day 9 and day 12 samples. Therefore, day 6 was considered the most representative time point for capturing key metabolic changes associated with the treatment.

6. Line 177: The thermo instruments should be labelled with their model and the country of manufacture.

Response: Thank you for your valuable comment. We have now specified the model and country of manufacture for the thermodynamic instrument at line 187 in the revised manuscript to ensure completeness and compliance with journal requirements. We appreciate your attention to detail, which has helped improve the clarity and rigor of our work.

7. The cDNA reverse transcription step should be included on line 188.

Response: We thank the reviewer for the correction. We have now supplemented the description of the cDNA reverse transcription step at line 203.

8. The use of the internal reference gene should be supplemented on line 196.

Response: We thank the reviewer for the suggestion. We have now included the information regarding the internal reference gene used at line 210.

9. What is the explanation for the build-up of anthocyanins at higher temperatures, yet the observation that the effect at 37 °C is less marked than at 25 °C?

Response: We thank the reviewer for this comment. Regarding the question of why anthocyanin accumulation was less pronounced at 37 °C compared to 25 °C, we have provided some analysis in the Discussion section. We attribute this phenomenon primarily to the thermal inactivation of anthocyanin biosynthetic enzymes. The biosynthesis of anthocyanins requires the involvement of a series of enzymes, such as phenylalanine ammonia-lyase (PAL). The optimal temperature range for most of these synthetases is around 25 °C, where their catalytic efficiency is highest. When the temperature rises to 37 °C, it approaches or exceeds the thermal inactivation threshold for some key enzymes, leading to a decline in the overall biosynthesis rate.

1. Jaakola, L., New insights into the regulation of anthocyanin biosynthesis in fruits. Trends in plant science, 2013. 18(9): p. 477-483.

10. In conclusion, line 545 is best applied to edible lilies.

Response: We thank the reviewer for the valuable suggestion. In response, we have revised the manuscript accordingly, with the modification clearly indicated at line 650. Considering the context involving postharvest technologies, we determined that using "edible lily" more accurately reflects the specific characteristics of the plant material studied compared to the more general term "lily." Therefore, we have replaced "lily" with "edible lily" in the text. We appreciate the reviewer's help in enhancing the precision of our manuscript.

---

## [Editor Report · Decision Letter 1]

6 Apr 2026

Light Quality-Regulated Anthocyanin Biosynthesis in Lilium leichtlinii subsp.  maximowiczii  Bulbs: A Multi-Omics Perspective

PONE-D-25-49782R1

Dear Dr. Wang,

We’re pleased to inform you that your manuscript has been judged scientifically suitable for publication and will be formally accepted for publication once it meets all outstanding technical requirements.

An invoice will be generated when your article is formally accepted. Please note, if your institution has a publishing partnership with PLOS and your article meets the relevant criteria, all or part of your publication costs will be covered. Please make sure your user information is up-to-date by logging into Editorial Manager at Editorial Manager® and clicking the ‘Update My Information' link at the top of the page. For questions related to billing, please contact  and clicking the ‘Update My Information' link at the top of the page. For questions related to billing, please contact billing support..

Kind regards,

Branislav T. Šiler, Ph.D.

Academic Editor

PLOS One
---

## [Editor Report · Acceptance letter]

PONE-D-25-49782R1

PLOS One

Dear Dr. Wang,

I'm pleased to inform you that your manuscript has been deemed suitable for publication in PLOS One. Congratulations! Your manuscript is now being handed over to our production team.

Kind regards,

on behalf of

Dr. Branislav T. Šiler

Academic Editor

PLOS One